



# Innovatory rainfall simulator design – A concept of moving storm automation

Ravi Kumar Meena[1], Sumit Sen[1], Aliva Nanda[1], Bhargabnanda Dass[1], and Anurag Mishra[2]

[1]Department of Hydrology, Indian Institute of Technology, Roorkee, Uttarakhand- 247667, India
[2]DSI, LLC, Washington, USA

**Correspondence:** Sumit Sen (sumit.sen@hy.iitr.ac.in)

**Abstract.** We developed an advanced design programmable rainfall simulator (RS) to simulate a moving storm rainfall condition. The RS consists of an automated nozzle control system coupled with a pressure regulator mechanism for an operating range of 50 kPa to 180 kPa at a drop height of 2000 mm above the soil flume surface. Additionally, a programmable mobile application was developed to regulate all RS valves. Near natural rainfall conditions were simulated at varying spatial and
temporal resolutions in a controlled environment. A soil flume of 2500 mm × 1400 mm × 500 mm was fabricated to conduct different hydrological experiments. The flume was designed to record overland, subsurface, and base flows simultaneously. This study focused on a detailed analysis of moving storms and their impact on hydrograph characteristics. Experimental results showed a considerable difference in terms of time to peak (tp), peak discharge (Qp), and hydrograph recession for two different storm movement directions (upstream and downstream). Two multiple regression models indicate a statistically sig-
nificant relationship between the dependent variable (tp or Qp) and the independent variables (i.e. storm movement direction, storm velocity, and bed slope gradient) at a 5% level of significance. Further, the impact of these moving storm phenomena reduces with the increase in the storm movement velocity.

## 1    Introduction

Due to high variability in storm pattern, intensity, storm movement velocity, direction, and rainfall drop sizes, it is often
challenging to study rainfall characteristics and impacts on overland flow, subsurface flows, baseflows, and soil erosion at a watershed-scale (Singh, 1998). Thus, rainfall simulation is one such method that is cost-effective and is used to study hydrological processes under controlled rainfall conditions (Nanda et al., 2018). Rainfall simulation refers to the process of simulating rain in a confined area for a specific time at a controlled rate. The understanding derived from rainfall simulation experiments is useful in many scientific disciplines i.e. hydrology, biosystems-engineering, agronomy, and geomorphology being disciplines
with enormous scope Lima et al. (2003). Rainfall simulator (RS) is an efficient instrument that enables quick data collection and analyses of a wide range of processes and treatment measures based on the variants of simulation configuration as per the study's objectives (Silveira et al., 2017).

Rainfall simulation experiments were developed by the United States Soil Conservation Service (SCS) in the 1930s to measure erosion potential and infiltration capacity of the soils. These experiments were used to develop the Universal Soil



Loss Equation (USLE), which is employed in various hydrological simulation models (Hall, 1970). Important application of RS studies has been developing the Phosphorus index (P-index) for estimating the impact of poultry litter application on water quality in the United States (DeLaune et al., 2010; Sheridan et al., 2008; Sharpley and Kleinman, 2010). Further, RS has been employed in various environmental studies like mine reclamation, agricultural nutrient transport (Sheridan and Noske, 2007), and forest hydrology (Croke et al., 1999) for the last 40 years. Rainfall simulation experiments have evolved with time

from mere sprinkler systems to sophisticated computer-based electrical and hydraulic systems (Smith and Schreiber, 1993; Cai et al., 2012). Based on the mechanism of rain droplets formation, RS can also be classified, i.e., (a) drip formers (Romkens and Roth, 1977) and (b) pressurized nozzles. Drip formers are used for small plot area, and low-intensity rainfall studies whereas pressurized nozzles are used for large scale field studies (10 to 500 m$^2$) Hall (1970); Neibling et al. (1981); Swanson (2013).

The experimental observations of rainfall simulation research can be upscaled to the larger hillslope- and catchment scale.

One of the example is a study conducted by (Nasri et al., 2002) for simulating sediment loss of 158 ha catchment using results of 1 m$^2$ experimental plot. Similarly, Nolan et al. (1997) showed erodibility rates of 144 m$^2$ tillage systems using a 1 m$^2$ simulation plot. The use of RS has also started in urban hydrology recently. Lima and Singh (2002) used single nozzle RS for moving storm analysis to study pollutant build-up, wash-off processes, and urban water quality. These experiments helped understand the slope-soil properties and the effect of surface resistance (i.e., vegetation and micro-topography conditions) on

overland flow and infiltration processes. Moreover, RS can also be used to study the flow routing, sediment generation, and transportation at different scales, i.e., plot-scale (Nanda et al., 2018) to hillslope-scale (Hall, 1970).

The Spatio-temporal distribution of rainfall influences the overland flow characteristics (Lima and Singh, 2002; de Lima and Singh, 2003), in terms of time to peak (t$_p$) and peak discharge (Q$_p$). An important assumption underlying in most hydrological experimental methods is that the rainstorm reaches instantaneously over the catchment and remains steady over it (Singh,

1998). Thus, such hydrological studies ignore the effect of storm movement on catchment runoff response. The exclusion of the storm movement could lead to poor estimation of runoff peaks Wilson et al. (1979).

Most of the time, RS cannot simulate spatio-temporal variability of rainfall like natural rainfall events, and most rainfall simulation studies take rainfall intensity as a constant parameter for a particular area at an instant. Inferences from such studies may not be representative of the actual process underway. To simulate the spatio-temporal variability of rainfall for lab

experiments, we designed an advanced programmable RS. The developed simulator was used to conduct different test scenarios in two different slopes and three different velocity conditions for both upstream and downstream storm direction. Following scientific parameters are considered for efficiency evaluation of developed moving storm rainfall simulator.

1. Peak discharge (Q$_p$) should be higher for the storm moving upstream to downstream than the storm moving in the opposite direction.

2. Time to peak (t$_p$) of the downstream directional storm should be lower than the upstream directional storm.

Further, the hydrographs generated under two different directions of storm movement (upstream and downstream) on a fully saturated bed condition were analyzed for the given experimental configurations; (a) Two different slope conditions (2.5% &





5%); and (b) Three different storm movement velocities (2 m min$^{-1}$, 3 m min$^{-1}$ and 6 m min$^{-1}$). A multiple regression model was used to test the statistical significance of the relationship between storm direction and the hydrograph characteristics.

## 2 Materials and Methodology

This section provides a detailed description of the soil flume and moving storm design along with the circuit diagram of the Bluetooth module.

### 2.1 Structural design

The rainfall simulator (RS) used in this study was designed at the Department of Hydrology, IIT Roorkee, India. The schematic
diagram of RS is presented in Figure 1. The instrument consists of a 3 m x 2 m frame connected with a pipe attached to a header (supporting 11 nozzles) and a pressure gauge. The frame was supported by four telescopic legs of 6 m each (Figure 1). The rainfall regulating structure connects to a centrifugal pump capable for controlling the water pressure and lifting the water from a feeder tank. The main components of the simulator are the frame, header for nozzle mounting, nozzles, and pumping station. A feeder tank was located near the RS to maintain a sufficient water supply. Water pressure in the system was adjusted by a
pressure regulator, and a "shut-off" valve was used to apply back pressure at the outflow end of the simulator system. Another valve was used to facilitate the accurate control of water pressure to the nozzles.

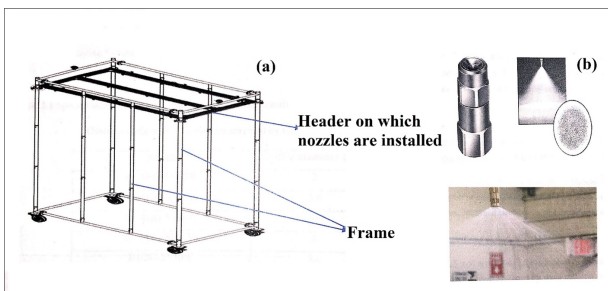

**Figure 1.** (a)Schematic diagram of RS. (b) Spraying System Co. Full jet G-style Spray Nozzle and Nozzle Spray flume. Source: Spraying System Co. Pvt Ltd, Bangalore.

Six full-cone nozzles manufactured by Spraying Systems Co. were used to simulate low to high-intensity rainfall. These nozzles produced a solid cone-shaped spray pattern with a circular impact area (Figure 1). A uniform spray coverage and distribution over a wide range of flow rates and pressure is possible using these nozzles (B1/88G-SS4.4W). The nozzles also
had removable caps and vanes for easy inspection and cleaning.

### 2.2 Design of soil flume

A 2.5 mm thick stainless-steel flume of 2500 mm × 1440 mm × 550 mm was fabricated to prepare the soil bed (Figure 2). A transparent acrylate wall at one vertical side of the soil flume facilitated easy visual observation. A base frame of 500 mm





height was designed for stability and to support the jack system. A manually operated worm wheel gear jack setup was installed
to change the slope of the flume (0 % - 7.5 %). The flume has three sub-partitions to accommodate three different soil types
at a single simulation. Outlets for surface flow, sub-surface flow, and base flow gauging were provided at the downstream end
of the flume. The surface and the subsurface flow outlets were placed at the height of 500 mm and 250 mm from the bottom
of the flume, respectively. The outlet for the baseflow measurement was located at the bottom edge of the flume. Additionally,
ten release/ seep slots (5 mm each) were provided at each sub-partition to analyze the change in the piezometric head. These
slots can also be used for leachate studies.

The soil flume was filled with gravel up to 50 mm depth to prevent the washout of the soil. Above gravel bed, sand was
added to a depth of 25 mm, and the remaining 425 mm flume space was filled with sandy loam soil (Figure 2b). The sand, silt,
and clay composition of the soil used was 66 %, 29 %, and 5 %, respectively, measured using a mechanical sieve analyzer.

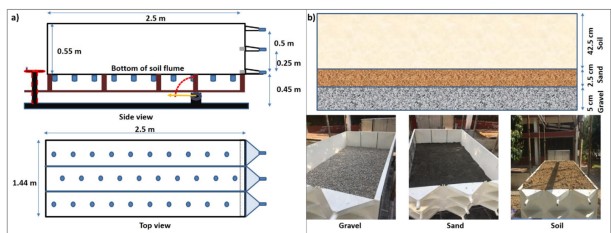

**Figure 2.** (a) Design of soil flume and (b)filling materials used in soil flume.

### *Moving storm design*

Detailed descriptions of components used to generate moving storm conditions are presented in Table 1 . A set of 11 nozzles
were used for simulating the moving storm condition. Electrically operated flow control valves were used to control these
nozzles through an Arduino Mega (AM) microcontroller board to simultaneously regulate the flow through nozzles. A nozzle
control system was incorporated using three components: servo-operated valve, AM microcontroller, and Bluetooth module
(BM) (Appendix A). This system serves two purposes; communication with the user interface in the handled device and control
of the opening and closing of nozzles. The detailed operational flowchart of the moving storm system is shown in Figure 3.
The nozzles were grouped into four clusters for this experiment to ease the flow regulation operation (Figure 4). Three clusters
(NC1, NC2, and NC3) consisted of three nozzles, and one cluster (NC4) consisted of two nozzles. These clusters were activated
and deactivated with a specific time gap to simulate the moving storm over the plot area. If needed, full control can be given
to each nozzle to regulate individually. To obtain variable rainfall intensities, a servo motor operated flow control valve was
inserted into the pipe openings just before the nozzle. Further, an android mobile application was developed for regulating the
valves through the BM.

The pressure regulating system (PRS) (Appendix B) comprises of a motorized globe valve, pressure transmitter, and Proportional-
integral-derivative controller. The PRS was designed to maintain constant pressure throughout the simulation to achieve a
constant rainfall intensity regardless of the opening and closing of the various number of nozzles.





**Table 1.** Specification of components used for moving storm rainfall simulator

| S.No. | Component | Specification | Qty. | Utility |
|---|---|---|---|---|
| 1 | Arduino | Mega | 1 | To control servo motors |
| 2 | Servo motor | MG995 | 11 | To operate valves |
| 3 | Bluetooth module | HC-05 | 1 | To receive signal from mobile |
| 4 | Valves | 15 mm | 11 | To control flow |
| 5 | Hexagonal nipple | 15 mm | 11 | Connect valves to main frame |
| 6 | Power supply | 12 V 1 A | 1 | Power supply for Arduino mega |
| | | 5 V 2 A | 4 | Power supply for servo |
| 7 | Motorized globe valve | 2 inches | 1 | To control bypass flow |
| 8 | Selec PID | PID500 | 1 | To control motorized globe valve |
| 9 | Pressure sensor | PT11 | 1 | To sense pressure in main line |

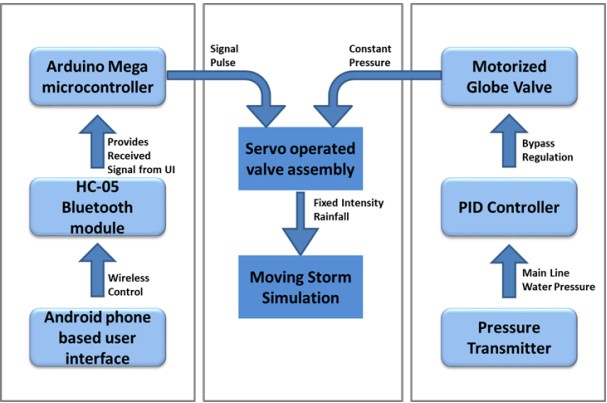

**Figure 3.** The operational structure of moving storm rainfall simulator.





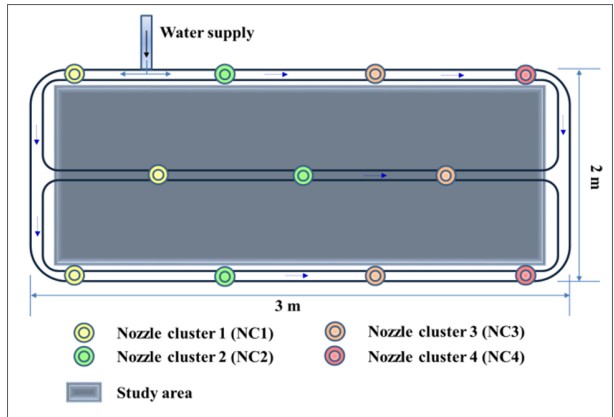

**Figure 4.** Nozzle distribution of rainfall simulator over the study area.

## 2.3 Circuit design

The circuit design comprised of AM, BM, and 11 servo motor-operated valves. The detailed design of the circuit diagram is given in Figure 5. The four connections, $R_x$, $T_x$, $V_{cc}$, and $G_{nd}$ in the BM were connected to pin 11, pin 10, 3.3 V, and $G_{nd}$ recipient pins in the AM, respectively. The 11 servos were clustered into four sets of 3, 3, 3, and 2 each in every cluster. The signal pins of these clusters of servo motors were connected to the digital signal pins of AM, numbered as pin 3, pin 5, pin 6 and pin 9. Each group had a power supply of 5 V – 2 A, and these were grounded to AM, which had a power supply of 12 V – 1 A.

The AM was coded (Appendix C) so that the servo motor could be regulated to control all the motors simultaneously, with any android based phone through a Bluetooth application. Two basic libraries used in this code were "SoftwareSerial.h' and "Servo.h." The software used to write the code is Arduino Editor, free coding software available online (https://auth.arduino.cc). The android OS application was designed to operate all of the 11 nozzles simultaneously, with one-touch Bluetooth connectivity and four slider bars to control four groups of servos operated valves at any value ranging from 0 to 100 percent. This application was developed using the "MIT app developer" software.

## 2.4 Simulation Uniformity Assessment

The following equation is used to calculate the Christiansen Uniformity Coefficient (UC)

$$\text{UC} = 100 \left[ 1 - \frac{\sum_{i=1}^{n} |X_i - \mu|}{\sum_{i=1}^{n} X_i} \right] \tag{1}$$

where $\mu$ is the average of all the measurements, $|X_i - \mu|$ is the sum of the individual deviations from the mean, and n is the number of measurements. The UC was measured using 66 beakers kept in a square array, 250 mm apart, beneath the simulator, covering the plot area of 2500 mm*1440 mm.

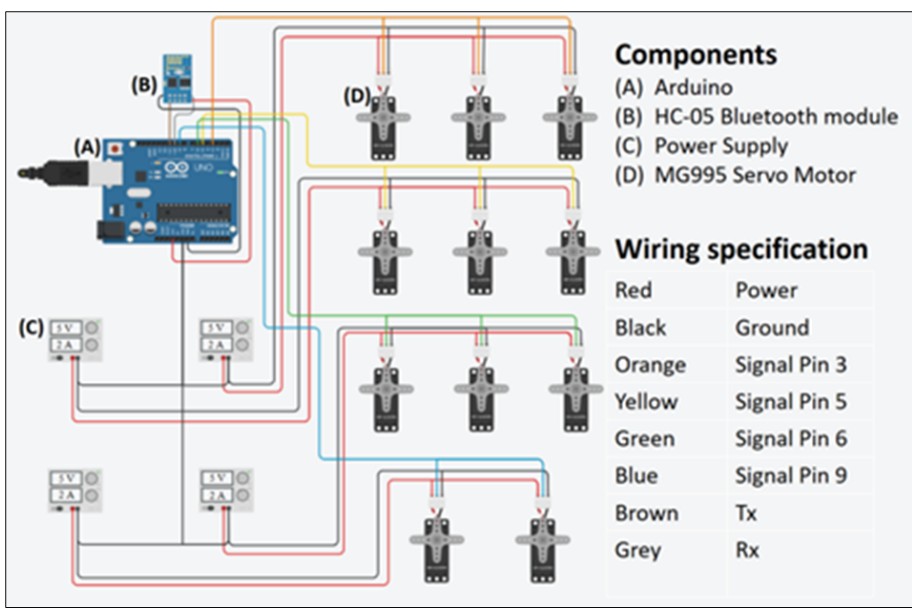

**Figure 5.** Circuit diagram of 11 servo motors operating through Arduino Mega and HC – 05 Bluetooth module.

## 2.5 Design of experimentation

The experiments consisted of 12 scenarios (Table 2 ) with three replications. Experiments were performed under fully saturated soil bed conditions to reduce the variability among scenarios and replications.

     The simulation results were analyzed using a multiple regression model considering one categorical variable (storm direction) and two numerical variables (velocity and bed slope) as independent variables and, time to peak ($t_p$) or peak discharge ($Q_p$) as the dependent variable. An indicator variable was used to include the direction of the storm in the multiple regression

model (downstream = 0, upstream=1). An indicator variable allows interpretation of the regression coefficients for storm direction as an additive effect on the hydrograph characteristics. By choosing the indicator variable for an upstream storm as 1, the regression coefficient of storm direction indicated the difference in the mean response of time to peak ($t_p$) or peak discharge ($Q_p$) of the upstream directional storm. A positive coefficient implies a positive effect upstream strom direction storm on the hydrograph characteristics compared to downstream directional storm. The null hypothesis of regression coefficient for each

of the independent variables as equal to zero was tested against alternative hypothesis of significant effect of the independent variable on storm hydrograph characteristics at 5% level of significance.

## 3   Results and discussion

After completing the design of the moving storm rainfall simulator (RS) (Figure 6), we checked the feasibility of the RS for generating the moving storm events. Before stepping into the different rainfall scenarios analysis, the rainfall distribution over





**Table 2.** Design structure of experiments

| Scenario | Slope (%) | Storm velocity (m min$^{-1}$) | Storm direction |
|---|---|---|---|
| 1 | 2.5 | 2 | Upstream |
| 2 | 2.5 | 2 | Downstream |
| 3 | 2.5 | 3 | Upstream |
| 4 | 2.5 | 3 | Downstream |
| 5 | 2.5 | 6 | Upstream |
| 6 | 2.5 | 6 | Downstream |
| 7 | 5 | 2 | Upstream |
| 8 | 5 | 2 | Downstream |
| 9 | 5 | 3 | Upstream |
| 10 | 5 | 3 | Downstream |
| 11 | 5 | 6 | Upstream |
| 12 | 5 | 6 | Downstream |

the plot was analyzed (Figure 7). It can be elucidated from the rainfall distribution graph that 70% of the plot area receives a uniform amount of rainfall, i.e., 30 mm to 36 mm. The lower rainfall (15 mm to 22 mm) amount was recorded at the plot edges. However, the overall UC was found to be 84.2 %. de Lima and Singh (2003) also conducted their rainfall simulator experiment with an average UC of 88 %.

### 3.1 Results of experimentation conducted at 2.5% slope

The results recorded by the rainfall simulator (RS) of the moving storm clearly exhibits the effect of storm direction, velocity, and slope on the overland flow hydrographs. For example, the hydrograph generated using the both downstream and upstream direction of storm movement with the velocity of 2 m min$^{-1}$ and 3 m min$^{-1}$ at a slope of 2.5% is shown in Figure 8. However, very little runoff was generated for velocity of 6 m min$^{-1}$, thus not included in the result.

The time to peak (t$_p$) of the hydrographs generated by the upstream to the downstream storm was less than the downstream 150 to the upstream storm irrespective of their velocities (Figure 8). When the storm moves towards the outlet (i.e., upstream to downstream), the overland flow was initiated at the upstream point and it moves downstream along with the storm direction. Thus, the collective overland flow reached the outlet simultaneously with the storm, which resulted in a sharp peak at the outlet of the soil flume. However, peak discharge (Q$_p$) of the upward moving storm was possible only when the entire catchment contributed, and that occurred when the storm reached the upstream point.

The hydrograph produced by the downstream to upstream storm movement displays a 110 s longer recession time compared to the opposite storm movement, especially during low storm velocity (2 m min$^{-1}$). The runoff and storm are in opposite direction





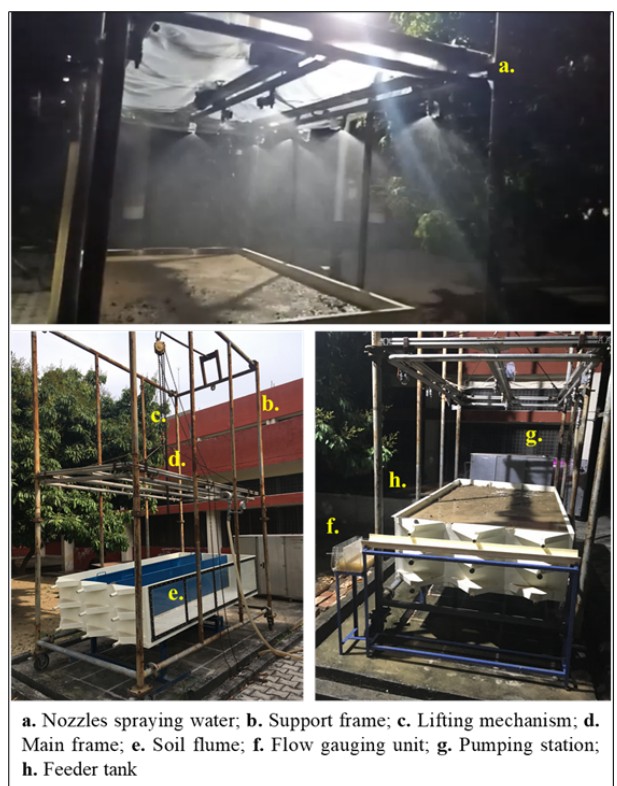

**a.** Nozzles spraying water; **b.** Support frame; **c.** Lifting mechanism; **d.** Main frame; **e.** Soil flume; **f.** Flow gauging unit; **g.** Pumping station; **h.** Feeder tank

**Figure 6.** Designed rainfall simulator for moving storm experiments.

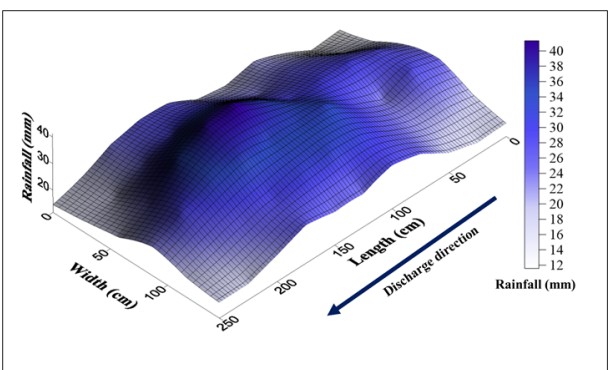

**Figure 7.** Rainfall distribution graph.

and thus, the upstream runoff water took longer time to reach at the outlet result in longer recession time. However, a negligible change in the recession time was observed between the storm directions during the 3 m min$^{-1}$ storm velocity (Figure 8b). The Lima and Singh (2003) also observed similar observations with an increase in storm velocity. When the storm was moving in





the upstream direction, 2 m min$^{-1}$ storm velocity showed a 150 s longer recession time than 3 m min$^{-1}$ velocity. But, no such change in recession time was observed between the velocity conditions for downstream directional storms.

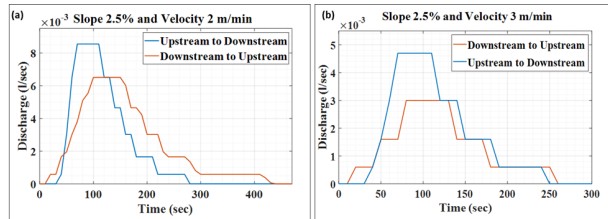

**Figure 8.** Hydrograph for velocity 2 m min$^{-1}$ (a), and 3 m min$^{-1}$ (b) at 2.5% slope.

### 3.2   Results of experimentation conducted at 5% slope

At 5% slope condition, rainfall simulation experiments were performed with three different velocities 2 m min$^{-1}$, 3 m min$^{-1}$, and 6 m min$^{-1}$ (Figure 9). The storm movement directions were the same as the previous experiments, i.e., upstream and

downstream. It can be clearly illustrated from Figure 9 that the recession characteristics, time to peak (t$_p$), and peak discharge (Q$_p$) followed the same trend as the hydrographs generated at 2.5% slope. An interesting observation was noticed during the testing of 6 m min$^{-1}$ storm velocity, i.e., the recession curve and Q$_p$ of both hydrographs completely matched with each other during the upstream and downstream directional storm movement (Figure 9c). Only t$_p$ varied slightly in these two hydrographs of 6m min$^{-1}$ velocity storm.

A detailed description of storm characteristics during different test scenarios is presented in Table 3 . It is observed that as the slope increased, the t$_p$ value decreased when the storm was moving to the upstream direction. While moving towards the downstream, the Q$_p$ value of 5% slope was 53.7 % and 43.3 % higher than 2.5% slope condition at a velocity of 2 m min$^{-1}$ and 3 m min$^{-1}$, respectively. Similarly, for upstream directional storm at a velocity of 2 m min$^{-1}$ and 3 m min$^{-1}$, the Q$_p$ of 5% slope was 59% and 42.8% higher than the 2.5% bed slope condition, respectively. These analyses concluded that the Q$_p$

value increases significantly with an increase in soil flume slope and decreases with increased storm velocity.

Further, observations show that the discharge volume followed the same relationship pattern as Q$_p$ with slope and storm velocity. The discharge volume of the 5% slope condition was 67.38% and 82 % higher than the 2.5% slope for the downstream and upstream directional storm, respectively. Similarly, for 3m min$^{-1}$ velocity, the increase in discharge volume due to slope increment was 58.56% and 55.56% for downstream and upstream storm directions, respectively.

From the above sets of experiments, it can be concluded that rainfall moving upstream results in; slower rise time, lower Q$_p$; and longer recession time (base time) compared to rainfall moving downstream. de Lima and Singh (2003) found that the high-velocity storm results in a smaller volume of runoff. This phenomenon was also observed in our study for both the slope conditions. For the 2.5 % slope condition, storm velocity of 6 m min$^{-1}$ barely generated any runoff at the plot outlet; thus, it did not create any runoff hydrographs. Moreover, de Lima and Singh (2003) observed that high storm velocity did not result

in significant changes in hydrographs during upstream and downstream storm movement because of the surface tension force.





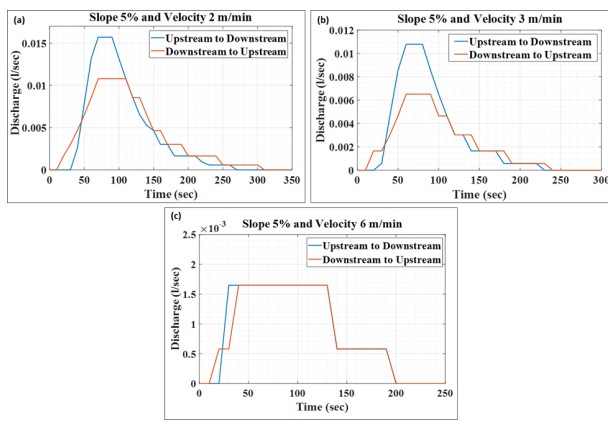

**Figure 9.** Hydrograph for velocity 2 m min-1 (a), 3 m min$^{-1}$ (b), and 6 m min$^{-1}$ (c) at 5% slope.

Similar observations are also noted in the present study during 6 m min -1 storm velocity (Figure 9c). The storm movement was so fast that the whole plot could never generate runoff at the outlet at an instant. The difference between $Q_p$ of the upstream

**Table 3.** Detailed moving storm characteristics of different scenarios

| Storm direction | Storm velocity (m/min) | Time to Peak ($t_p$)(sec) | Peak discharge ($Q_p$)(l/sec) |
|---|---|---|---|
| Slope 2.5 % | | | |
| Downstream | 2 | 70 | 0.0086 |
| Upstream | 2 | 100 | 0.0065 |
| Downstream | 3 | 70 | 0.0047 |
| Upstream | 3 | 80 | 0.0030 |
| Slope 5 % | | | |
| Downstream | 2 | 70 | 0.016 |
| Upstream | 2 | 70 | 0.011 |
| Downstream | 3 | 60 | 0.011 |
| Upstream | 3 | 60 | 0.007 |
| Downstream | 6 | 30 | 0.0016 |
| Upstream | 6 | 40 | 0.0016 |

and downstream directional storm is the function of storm velocity. Further, Lima et al. (2011) discussed the influence of slope on moving storm runoff hydrograph, i.e., how the runoff volume increases with an increase in slope angle. The effect of storm direction on $Q_p$ is also discussed by de Lima and Singh (2003), Isidoro et al. (2012) and Seo and Schmidt (2013).






They concluded that the downstream storm produces higher Qp than the upstream directional storm. Similar results were also obtained in the current study (Table 3 ).

To further test the significance of the effect of the experimental variables on the observed differences in time to peak ($t_p$) and peak discharge ($Q_p$), multiple regression analysis was performed. Regression analysis with an indicator variable for storm
direction results in two regression model equations for each hydrograph characteristic modelled (Eq.(2)).

$$T_p\,or\,Q_p = \begin{cases} \beta_0 + \beta_1 + (\beta_2 * velocity) + (\beta_3 * slope), \\ \quad when\ direction = upstream = 1 \\ \beta_0 + (\beta_2 * velocity) + (\beta_3 * slope), \\ \quad when\ direction = downstream = 0 \end{cases} \tag{2}$$

The model fit details for the regression models are shown in Table 4 . Interpreting the value of $\beta1$ in the models, for a storm moving upstream, $t_p$ is higher (positive) by $10 \pm 4.776$ sec and $Q_p$ is lower (negative) by $0.00256 \pm 0.00086\ \mathrm{l\ sec^{-1}}$. These values are significant at 5% level of significance thus provides a satisfactory hydrological verification for the moving
storm rainfall simulator. The multiple regression models for $t_p$ and $Q_p$ explain 89% and 94% of the variability in $t_p$ and $Q_p$ observations (as given by model R square values), thus indicating the sufficiency of the model for explaining the relationship between experimental variables and hydrograph characteristics. The sign of the regression coefficients of storm velocity and bed slope agree with the empirical conclusions that $Q_p$ increases with an increase in bed-slope and decrease in velocity. In contrast, both bed slope and velocity have a negative effect on $t_p$.

Table 4. Regression coefficients, standard error, the t-statistic and associated p-value for multiple regression models

| | Coefficients | Standard Error | t Stat | p-value |
|---|---|---|---|---|
| **Time to peak($t_p$)** | | | | |
| **Intercept** | 111.15 | 4.29 | 25.9 | <0.05 |
| **Direction** | 10.00 | 2.32 | 4.3 | <0.05 |
| **Velocity** | -8.36 | 0.86 | -9.7 | <0.05 |
| **Slope** | -6.10 | 1.03 | -5.9 | <0.05 |
| **Peak discharge ($Q_p$)** | | | | |
| **Intercept** | 0.00852 | 0.00077 | 11.0 | <0.05 |
| **Direction** | -0.00256 | 0.00042 | -6.1 | <0.05 |
| **Velocity** | -0.00290 | 0.00015 | -18.8 | <0.05 |
| **Slope** | 0.00229 | 0.00018 | 12.3 | <0.05 |

The main objective of this study is to develop a handy, multifunctional, and advanced rainfall simulator to study the moving storm rainfall pattern. After development, 36 experiments characterized into 12 different scenarios were conducted to check the instrument's feasibility. From the above discussion, it is evident that the tool could generate the moving storm condition

satisfactorily. However, this study was limited to evaluating the impact of storm movement on generated hydrograph under a single storm pattern with two different slope conditions, two different storm directions, and three different velocities. This

advance designed RS can be considered to be used in the future to analyze the impact of storm movement over soil erosion and nutrient transport. Further, the designed flume can be used for subsurface flow, base flow, and leachate studies.

## 4   Conclusions

A multi-nozzle programmable RS was designed to simulate moving storm rainfall. 12 different test scenarios were examined in two different slopes (2.5% & 5%) conditions and three different velocities (2 m min$^{-1}$, 3 m min$^{-1}$ & 6 m min$^{-1}$). In these

experiments, storm movement was considered along the slope (downstream) and against the slope (upstream) of the basin, keeping rainfall intensity and soil saturation constant. The results indicate a statistically significant influence of spatial and temporal distributions of rainfall on hydrograph and its characteristics.

Following conclusions were drawn from the study:

1. The hydrograph generated from the downstream moving storms yielded a higher peak with a sharp rise and short reces-
sion limb.

2. The upstream moving storm produced a runoff hydrograph of lower peak and a prolonged, gradually decreasing recession limb.

3. With the increase in the storm movement velocity, the impact of moving storm direction in terms of discharge peak and time to peak became negligible in either direction of storm movement.

4. An increase in the basin slope reduced the impact of moving storm direction on overland flow.

*Code availability.*  The software code is available in GitHub https://github.com/rmeena64/ravi/blob/7b9938c2a86e18e69adecedc3c5e95557b28b94b/ program_code.rtf.

*Data availability.*  Experimental data can be obtained from corresponding author upon reasonable request.

## Appendix A:  Nozzle Control System

### A1   Servo operated valve

A stop cock valve was used to develop a servo-operated valve due to its low operational torque requirement. Servo motor of torque 10 kg cm$^{-2}$ which is easy to control and have fairly high accuracy was used to control the valve. An aluminum frame was fabricated using a 2.5 mm aluminum sheet to hold the servo motor and the stop cock valve together.

Servo motor MG995 is a High-Speed Digital Motor with a rotation angle of 90° in both directions enabling a 180° reach.
Pulse Width Modulation signals were used for the operational control of the servo motor to increase process speed and efficiency. It was equipped with an internal circuit which gave high torque and better stability.

## A2   Arduino mega

Arduino Mega (AM) is a microcontroller board with fifty-four digital I/O pins, four hardware ports, sixteen analog ports and a 16 MHz crystal oscillator complemented by an ICSP header, a power jack and a reboot button. It can be powered through both
USB as well a DC supply of 7-12 V 1A[21].

## A3   Bluetooth module

A HC-05 Bluetooth Module (BM) can be enabled both as a Master, and as a slave. The Master setting enabled auto-communication between the two Bluetooth devices whereas the slave set could only accept the incoming connection from Master Bluetooth device. It had a 3 Mbps data transmission speed with a 2.4 Giga hertz transmitter and receiver. It is comprised of six pins, $V_{cc}$
– for power supply; $G_{nd}$ – for negative; $T_x$ – for transmission; $R_x$ – for receiving; a Key to switch between Master and Slave and an LED to display its operational activity[22]. In this experiment, the Bluetooth module was used as a slave set (default configuration).

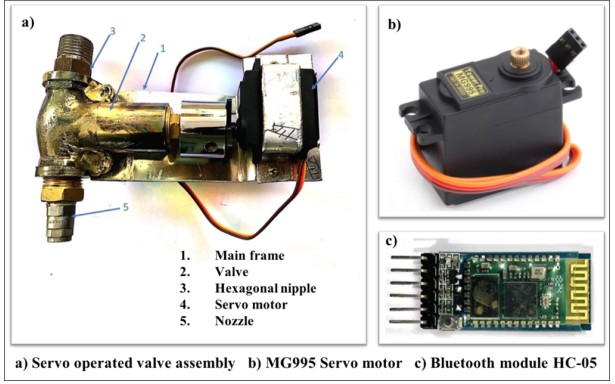

**Figure A1.** Components used for nozzle control system.

## Appendix B:  Pressure Regulating System (PRS)

## B1   Motorized globe valve

A motorized globe valve was used to control the bypass flow and maintain a constant pressure in the main line which worked through a pressure feedback circuit from the main line. A 2-way globe valve of metal to metal seating with a pulse based modulation of 4 – 20 mA actuator was used. It had an intelligent circuit that sensed hindrance in valve movements. An AC





sensor was used for circuit protection that can shut down the valve during an overload condition. It had a robust and compact design to ease the installation.

## B2 Pressure transmitter


A Mass PT11 pressure transmitter has a very compact design with stainless steel construction. It is highly stable against shock and vibration and also have features such as reverse polarity, limit protection and have high accuracy. This pressure sensor was installed to check the main line pressure. PID controller can sense the change in pressure and can act accordingly to operate bypass and to maintain a constant pressure in the main line for a uniform rainfall intensity.

## B3 Proportional-integral-derivative (PID) controller


Selec PID500 is a controller which is employed widely in the industrial process controls. PID controller is a control loop feedback system and is used to operate a motorized globe valve on the basis of an input signal from a pressure sensor. Whenever there is a pressure offset from the set value, it sends a signal to the motorized valve to re-attain the set value. The I/O signal from the pressure sensor and PID respectively ranged between 4-20 mA. The controller had a compact square housing with panel mounting facility in its enclosure, powered by a 240 V AC supply[23]. PID controller was used to control the bypass flow by operating a motorised valve to maintain constant pressure in the main line against any pressure drop generated due to the moving storm simulation


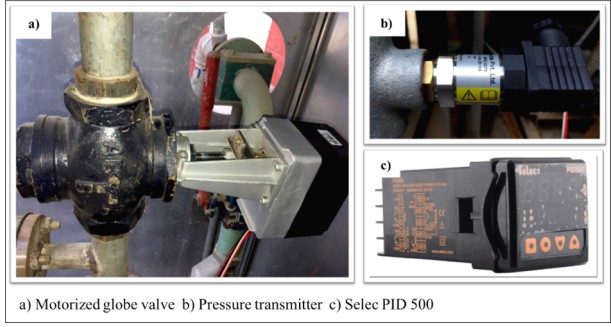

a) Motorized globe valve  b) Pressure transmitter  c) Selec PID 500

**Figure B1.** Components used for pressure regulating system.

*Author contributions.* The main contributions from each co-author are as follows. RKM contributed to the methodology, software, validation, and draft preparation. SS contributed to the conceptualization, supervision, and funding acquisition. AN and BD contributed to the visualization, writing, and editing of the paper. AM contributed to the review of paper.


*Competing interests.* The authors declare that they have no conflict of interest.



*Acknowledgements.* The authors would like to acknowledge the Department of Hydrology, Indian Institute of Technology Roorkee, Roorkee, India, for providing support and resources for this experimental study. The authors would also like to acknowledge Prof. Ramesh A. and Mr. Denzil Daniel for statistical analysis interpretation and Dr. Prasanthi Pallapu for proofreading the initial draft of the manuscript.

*Financial support.* The research has been supported by Department of Hydrology, Indian Institute of Technology Roorkee, India.



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
