# Peer review of "A contribution to rainfall simulator design – A concept of moving storm automation"

_Hydrology and Earth System Sciences, 2021_

## Referee Comment (RC1)

**General comment**

The manuscript analyzes the impact of a moving storm rainfall condition on the hydrograph characteristics using a programmable rainfall simulator. The authors present the results of different rainfall simulation experiments on the two hydrograph characteristics, the time to peak, and the peak discharge. The experiments were performed to test different scenarios in two different slopes and three different velocity conditions for both upstream and downstream storm direction.

This topic is very interesting, and the developed rainfall simulator system is very innovative. However, I think that the paper could be improved by the authors that would make it more attractive. Some aspects, mainly related to the developed rainfall simulator system are not clear, and others are not presented. The characteristics of the simulated rainfall are not well described, and the test to verify if it well reproduces the natural rainfall are not discussed. Those aspects are very important when using a rainfall simulator system.

1.The rainfall simulation experiments were used to evaluate the peak discharge and the time to peak flow. Those two parameters are more influenced by the physiographic properties of the watershed, vegetation cover, flow dynamics, and soil water content at the begin of the rainfall event. Considering that the system includes a soil flume, why the authors didn't use it to simulate different soil water content conditions, for example?

2. Why did you not include the system description in the main text? In my opinion, the mechanical innovations presented for this rainfall simulator is one of the most important part in the paper

3. Did you analyzed the characteristics of the simulated rainfall? Is the system able to simulate a natural rainfall?

**Specific comments**

[L.25] More recent rainfall simulators have been developed but are not mentioned in the manuscript. Please add more recent references.

[L.64] The system description is not complete. What is the high of the system? Which intensities is the systems able to simulate? How did you verify if the system reproduces a real rainfall event? Did you verify the raindrops distribution? The characteristic of the simulated rainfall should be verified using a disdrometer, for example. The authors say that the system simulated "Near natural rainfall conditions" (abstract L.6], but I don't know what it means. The raindrop distribution should be investigated for all the simulated area.

[L.104] What is the system pression? What is the intensity of the simulated rainfall?

[L.112] What is "Appendix C"? It is not presented in the paper.

[L.120] What are the values for the coefficient of uniformity? Did you test the uniformity of the simulated rainfall for different rainfall intensities?

[L.125] How did you considered the wind effect during rainfall simulations? Is there any protection?

[L.125] Is there any change of the soil surface after each experiment? Is there any change of the roughness, for example?

[L.141] What is the mean intensity of the simulated rainfall? Why are you simulating this intensity?

[L.147] How did you chose the rainstorm movement velocity?

[L.133] Please change "strom" to storm.

[L.288]. check the reference: The style is not correct.

---

## Author Response (AR1)

**Reply to Editor**

**General comments**

 The paper discusses the development and preliminary testing of a pressurised rainfall simulator. The simulator consists of an automated nozzle control system coupled to a pressure regulator mechanism, allowing it to automatically control rainfall intensity and simulate storm movement. The preliminary tests were carried out with a soil flume. I would like to congratulate the authors for this very interesting manuscript, which is a step further in rainfall simulation. However, there are some flaws – in my opinion – that could/should be addressed for the sake of easiness of reading and scientific soundness. Figures and tables, however, are clear to understand and useful, as they are a key-element to understand the authors' point-of-view. There are some issues regarding scientific soundness or, at least, lack of clarity. Some of these issues are referred to below in the "specific comments".

**Reply:** Authors are thankful to the editor for providing very insightful comments and suggestions. We have incorporated all the specific and technical comments in our revised manuscript.**

2. Referred literature is relevant but it lacks recently published information on rainfall simulators development. Some citations seem to be casuistic, and I've only noticed one reference (de Lima and Singh, 2003) to be strongly discussed/compared with this paper's findings. Moreover, there are some flaws in the references list.

Reply: We have added some recent references (not specific to moving storm rainfall simulation) in introduction (second para, second last line) and in the result and discussion section (first para while discussing uniformity coefficient).

**From results and discussion section:**

"The lower rainfall (15 mm to 22 mm) amount was recorded at the plot edges. However, the overall 180 UC was found to be 84.2 %. de Lima and Singh (2003) also conducted their rainfall simulator experiment with an average UC of 88 %. Similarly, Macedo et al. (2021), and Salem and Meselhy (2021) conducted rainfall simulation experiments for studying soil erosion at a UC of 75 %, and 89- 94 %, respectively. Further, Mendes et al. (2021) also carried out simulation tests for studying geotechnical and hydrological phenomena with a UC of 75 %."

3. Apart from research, rainfall simulators can be useful tools for visualisation and pedagogical purposes. Moreover, this paper presents an important advance in rainfall simulation. Two different parts can be identified in the paper: 1) the description of the device itself (construction of rainfall simulator, electronics, coding, operational control, ...), and the preliminary tests conducted (use of a soil flume, rainfall intensity uniformity assessment, analysis of surface runoff hydrographs, ...). My major criticism to the paper is that I find these two parts to be someway confused along the text, i.e., despite the quality of the English being very good I did not like the way the paper is organised.

**Reply:** Thank you for highlighting the major scientific focus of our manuscript. We have changed the flow of our manuscript (moving appendix to main text). We hope that would be easy for readers to get the essence of the manuscript.

**General Comments**

Q1 – A soil flume was used, with the capability to gauge surface flow, sub-surface flow, and baseflow. However, only surface flow hydrographs were presented. Why? If these flows were not to be analysed and discussed, why is this detail about the soil flume presented? I would suggest, at least, to clearly state in Section 2.2 (Design of soil flume) that only the surface flow data is analysed in the paper.

Reply: Thank you for the suggestions. We have designed the soil flume with future scope for multiple studies. The main attraction of the manuscript is the design of a moving storm rainfall simulator along with the multi-functional soil flume. So, we discussed the complete design detail of the soil flume. However, in the current study, we evaluated the moving storm rainfall conditions using the developed experimental setup. We added in section 2.2 (last line of the first paragraph) about the usage of soil flume in this study.

Q2 – Why was a soil flume used? If the paper is (supposed to be) focused on the rainfall simulator, why did not the authors use a much simpler impervious surface?

Reply: Initially, we were planning to study soil erosion using the rainfall simulator but we were not able to acquire the laser rainfall analyzer for measuring drop size distribution and terminal velocity (due to financial constraints). Thus, we limit the study to the general testing of a moving storm rainfall simulator.

Q3 - Why did the authors present the very interesting electronic control system(s) as appendixes? This is the main novelty of the paper! There are many papers regarding rainfall simulation. However, there are no papers regarding rainfall simulators with the capabilities and automatisation of this one.

**Reply:** Thank you for your suggestion. We have added these sections (section 2.3 and section 2.4) in the paper instead of appendixes.

Q4 – During the simulated rainfall experiments, which were the criteria to consider the beginning and the end of discharge?

**Reply:** We used the beginning of the storm simulation as the beginning of the discharge measurement, and we measured the discharge from the soil flume stopped completely.

**Specific comments**

[Title] "Innovatory [...]" is ambiguous... maybe something like "A contribution to [...]" could sound better.

**Reply: Thank you for the suggestions.**

[P.1; L.4] "Near natural rainfall conditions". What do the authors mean by this? And how can you assure that the artificial rainfall produced by this novel rainfall simulator is similar to natural rainfall? There is no raindrop analysis (e.g., drop spectra analysis), and the only analysis of rainfall characteristics regards the rainfall intensity spatial uniformity. In my opinion, the authors cannot assure that the simulator produces "Near natural rainfall conditions", at least by the information provided in the paper.

Reply: Thank you for your suggestion. We have used "near-natural rainfall conditions" in terms of the storm movement as most of the previously developed rainfall simulators just produce a still rainfall as we study rainfall in theory.

[P.2; L.26] Why is estimating the impact of poultry litter application on water quality of particular importance? I am not saying it is not important, but for sure I would think of other uses for a rainfall simulator first, such as soil erosion (after all, the authors used a soil flume...) or drainage/flood simulation. This is an example of what I find to be a casuistic citation, as I cannot find anything else on the paper minimally related to poultry litter application.

**Reply: We removed this statement.**

[P.2; L.32-33] The authors state the "Drip formers are used for small plot area, and low-intensity rainfall studies whereas pressurized nozzles are used for large scale field studies (10 to 500 m2)". However, this pressurised rainfall simulator is to be used with plots smaller than 10 m2. Can the authors comment on this?

Reply: The mentioned statement is from two old papers of Romkens and Roth, 1977 and Hall, 1970. However, in the recent papers, researchers used pressurized nozzles for the flume area of  $0.9 \text{ m}^2$  to 7.5 m2 (de Lima and Singh, 2003; de Lima et al., 2009; de Lima et al., 2011; Isidoro et al., 2011 and Isidoro et al., 2013). To avoid confusion, we have removed these statements from the manuscript.

**References:**

- 1. De Lima, J. L. M. P., & Singh, V. P. (2003). Laboratory experiments on the influence of storm movement on overland flow. Physics and Chemistry of the Earth, Parts A/B/C, 28(6-7), 277-282.
- de Lima, J. L. M. P., Dinis, P. A., Souza, C. S., De Lima, M. I. P., Cunha, P. P., Azevedo, J. M., ... & Abreu, J. M. (2011). Patterns of grain-size temporal variation of sediment transported by overland flow associated with moving storms: interpreting soil flume experiments. Natural Hazards and Earth System Sciences, 11(9), 2605-2615.
- 3. De Lima, J. L. M. P., Tavares, P., Singh, V. P., & de Lima, M. I. P. (2009). Investigating the nonlinear response of soil loss to storm direction using a circular soil flume. Geoderma, 152(1-2), 9-15.
- 4. Isidoro, J. M., de Lima, J. L., & Leandro, J. (2012). Influence of wind-driven rain on the rainfall-runoff process for urban areas: Scale model of high-rise buildings. Urban Water Journal, 9(3), 199-210.
- 5. Isidoro, J. M., de Lima, J. L., & Leandro, J. (2013). The study of rooftop connectivity on the rainfallrunoff process by means of a rainfall simulator and a physical model. Zeitschrift für Geomorphologie, Supplementary Issues, 177-191.

[P.2; L.53] It is not clear how the "efficiency evaluation" is performed.

**Reply: We reworded the statement to the only "evaluation" which means evaluation of moving storm rainfall simulator using different characteristics of surface runoff hydrograph (i.e., peak discharge and time to peak).**

[P.3; L.73] I do not agree with stating that "A uniform spray coverage [...]", as the spatial rainfall distribution is (factually) not uniform. The CUC analysis is presented only in Section 3 (Results and discussion), and just for one rainfall intensity scenario. This is by no means enough to state that this rainfall simulator can produce uniform spray coverage... in fact, one of the major problems of pressurised rainfall simulators. The paper does not prove that this rainfall simulator can produce uniform spray coverage.

Reply: We did the uniformity test for different rainfall intensities (at multiple rainfall intensities between the range of 36 mm/h to 606 mm/h with the minimum UC of 82 % and maximum UC of 91 %). However, in the current manuscript, we only mentioned the uniformity coefficient of the intensity used for this study.

[P.4; L.81] Base flow is incorrectly used here. It should be "groundwater flow" or "deep sub-surface flow" (the latter is better). "Baseflow" is the part of streamflow that is sustained between precipitation events, and that flows to streams by delayed pathways. It has nothing to do with the flow physics detailed in this paper.

**Reply: Totally agreed with your point. We have corrected the statement.**

[P.4; L.103-104] Did the use of flexible hoses to supply water from the feeder tank to the nozzles resulted in difficulties to maintain a steady pressures, mainly when opening/closing the valves? I suggest looking at Isidoro and de Lima (2015) and comparing the advantages/disadvantages of this novel system regarding pressure stabilisation.

*Hydraulic system to ensure constant rainfall intensity (overtime) when using nozzle rainfall simulators. Hydrology Research (2015) 46 (5): 705–710. DOI: 10.2166/nh.2015.087*

Reply: Thank you for your suggestion. We have used flexible hoses to supply water from the feeder tank to the header. In this rainfall simulator design, there continuous opening and closing of the nozzle happens to simulate a moving storm which will lead to change in system pressure. To compensate the impact of change in pressure, we used feedback system which continuously checks the system pressure and according to the condition it maintains the bypass flow (return flow) of the system to keep the pressure constant.

[P.8; L.147-148] "However, very little runoff was generated for velocity of 6 m min-1 [...]". Is this true both for the upstream and downstream storm movement tested scenarios?

**Reply:** Yes, for both upstream and downstream directions the generated runoff was not sufficient to generate a hydrograph.**

[P.13; L.231] Please explain better "A stop cock valve was used to develop a servo-operated valve due to its low operational torque requirement".

**Reply:** Stop cock valves used in this study are quarter-turn valves which require very less operation force as compared to the other types of valves such as ball valve, butterfly valve, gate valve.

[P.13; L.232] Please check this. Torque unit (SI) is Nm.

Reply: Thank you for your suggestion. That was a typo error too the initially unit of torque on the specification sheet for the used servo motor was 10 kg/cm but now we converted it and changed it to 0.98 Nm (currently in section 2.3.1).

[P.14; L.250] What do the authors mean by "bypass flow"? Please detail this further.

**Reply:** Thank you for your suggestion. In this rainfall simulator, the header of the rainfall simulator is closed so the pressure of the simulator line is maintained by controlling the bypass flow (return flow) (we added in section 2.4.1).

**Technical / typos / orthography comments**

[P.2; L.52-53] I suggest using "The following parameters are considered [...]" instead of "Following scientific parameters are considered [...]"

**Reply: Corrected**

[P.4; Figure 2 caption] I suggest adding "(not in scale)" to this figure's caption.

**Reply: added**

[P.4; L.91-92] Please check this sentence.

**Reply:** We have changed the statement as following: "A set of 11 nozzles were used for simulating the moving storm condition. Electrically operated flow control valves were used to control these nozzles through an Arduino Mega (AM) microcontroller board."

[P.4; L93-94] I suggest using "Bluetooth Module (BM)" instead of "Bluetooth module (BM)".

**Reply: Corrected**

[P.4; L.102] I suggest using "Pressure Regulating System (PRS)" instead of "pressure regulating system (PRS)".

**Reply: Corrected**

[P.4; L.102-103] I suggest using "Proportional-integral-derivative controller (PID controller)" for ease of understanding.

**Reply: Corrected**

[P.5; Table 1] Why are the nozzles not listed on this table?

**Reply: Thank you for your suggestion. We have added that to the table.**

[P.6; Figure 4 + caption] I suggest using "soil flume" instead of "study area".

**Reply: Corrected**

[P.6; L.112] Appendix C is missing on the paper. However, L.226 (P.13) shows a link for the software code. Is this the code supposed to be presented in appendix C?

**Reply: Thank you for pointing that out. Yes, we have already added the link of the code used in this study but due to the typo error we missed to put it as the Appendix C. That correction has been made in the manuscript.**

[P.9; L.158-159] There is an error in the citation... it should be "de Lima and Singh (2003)".

**Reply: Thank you for your suggestion. We will change that to the manuscript.**

[P.10; L.169] Please use "6 m" instead of "6m".

**Reply: Corrected**

[P.10; L.177] Is it 82.00 % ? (All other values show two decimal places).

**Reply: Corrected**

[P.10; L.178] Please use "3 m" instead of "3m".

**Reply: Corrected**

[P.11; L.187] I suggest using "[...] could never contribute to generating runoff [...]" instead of "[...] could never generate runoff [...]".

**Reply: Corrected**

[P.11; Table 3] I suggest using  $\times 10^{-3}$  in the table's last column.

**Reply: Incorporated the suggestion**

[P.13; L.209] I suggest using "[...] and three different moving storm velocities" instead of "[...] and three different velocities".

**Reply: Corrected**

[P.13; L.214] I suggest using "[...] and three different moving storm velocities" instead of "[...] and three different velocities".

**Reply: Corrected**

[P.15; L.265] Please check this line of text where a reference (?) [23] is incorrectly presented.

**Reply:** This is a typo error. There were two more typos in line L.240 and L.246 (submitted version). We removed these typos (currently in sections 2.3.2, 2.3.3 and 2.4.3).**

[P.17; L.288] Please check this reference. The first author's last name is "de Lima". (It is correctly presented in L. 281).

**Reply: Thank you for pointing this out. We have incorporated the suggestions.**

[P.17; L.290] Please check this reference. The first author's last name is "de Lima". (It is correctly presented in L. 281).

**Reply: Thank you for pointing this out. We have incorporated the suggestions.**

**Reply to Reviewer 1**

The manuscript analyzes the impact of a moving storm rainfall condition on the hydrograph characteristics using a programmable rainfall simulator. The authors present the results of different rainfall simulation experiments on the two hydrograph characteristics, the time to peak, and the peak discharge. The experiments were performed to test different scenarios in two different slopes and three different velocity conditions for both upstream and downstream storm direction.

This topic is very interesting, and the developed rainfall simulator system is very innovative. However, I think that the paper could be improved by the authors that would make it more attractive. Some aspects, mainly related to the developed rainfall simulator system are not clear, and others are not presented. The characteristics of the simulated rainfall are not well described, and the test to verify if it well reproduces the natural rainfall are not discussed. Those aspects are very important when using a rainfall simulator system.

Reply: We are thankful to the reviewer for giving constructive feedback. We have addressed all the comments as best as possible in our manuscript. Further, we also want to mention that the details about the rainfall characteristics and the reasoning behind the selection of that intensity have been added in the first paragraph of the results and discussion section.

1. The rainfall simulation experiments were used to evaluate the peak discharge and the time to peak flow. Those two parameters are more influenced by the physiographic properties of the watershed, vegetation cover, flow dynamics, and soil water content at the begin of the rainfall event. Considering that the system includes a soil flume, why the authors didn't use it to simulate different soil water content conditions, for example?

Reply: Thank you for your suggestion. Initially, we also planned to work with a specific moisture content of the soil but it very complicated to keep the moisture content constant for each simulation, and the sole purpose of the data presented in this manuscript is to show that the designed rainfall simulator is capable of simulation moving storm with different storm movement velocities.

2. Why did you not include the system description in the main text? In my opinion, the mechanical innovations presented for this rainfall simulator is one of the most important part in the paper

**Reply: Thank you for your suggestion. We have added these sections in the paper instead of appendixes.**

3. Did you analyze the characteristics of the simulated rainfall? Is the system able to simulate a natural rainfall?

Reply: The maximum 3-min (simulated) rainfall intensity is 40 mm/h. The drop size distribution and kinetic energy of rainfall are not measured as we don't have a laser sensor. In terms of storm movement, yes, the simulator does have the capability to generate near-natural rainfall conditions. Further, we want to mention that the maximum 5-min rainfall intensity varies in the range of 12 mm/h to 109 mm/h in our Aglar experimental watershed, Uttarakhand, India (Nanda et al.,2019). So, the used intensity seems reasonable. We have mentioned same in first paragraph of Section 3.

Nanda, A., Sen, S., & McNamara, J. P. (2019). How spatiotemporal variation of soil moisture can explain hydrological connectivity of infiltration-excess dominated hillslope: Observations from lesser Himalayan landscape. Journal of Hydrology, 579, 124146.

**Specific comments**

[L.25] More recent rainfall simulators have been developed but are not mentioned in the manuscript. Please add more recent references.

Reply: Thank you for your suggestion. We did not include the recent designs because as per best of our knowledge most of the new rainfall simulator designs have a goal of portability and spray uniformity but those are not designed to simulate moving storms. However, we have added some recent references of portable rainfall simulators discussing about their uniformity coefficient (Section 3, first para).

[L.64] The system description is not complete. What is the high of the system? Which intensities is the systems able to simulate? How did you verify if the system reproduces a real rainfall event? Did you verify the raindrops distribution? The characteristic of the simulated rainfall should be verified using a disdrometer, for example. The authors say that the system simulated "Near natural rainfall conditions" (abstract L.6], but I don't know what it means. The raindrop distribution should be investigated for all the simulated area.

Reply: Thank you for your suggestion. We have added the relevant details as you mentioned in the above comment. Initially, we were planning to do the other studies too relevant to soil erosion but we were not able to acquire the laser rainfall analyzer for drop size distribution and terminal velocity, so we limited the study to general testing of moving storm rainfall simulator

[L.104] What is the system precision? What is the intensity of the simulated rainfall?

Reply: System pressure of the rainfall simulator at the time of simulation is 0.6 kg/cm2 to simulate a rainfall with a mean intensity of 36 mm/h.

[L.112] What is "Appendix C"? It is not presented in the paper.

**Reply:** Thank you for pointing that out. We have already added the link of the code used in this study and Appendix C is a typo error. That correction has been made in the paper.**

L.120] What are the values for the coefficient of uniformity? Did you test the uniformity of the simulated rainfall for different rainfall intensities?

Reply: We did the uniformity test for different rainfall intensities (at multiple points between the range of 36 mm/h to 606 mm/h with the minimum UC of 82 % and maximum UC of 91 %) but we only mentioned the uniformity coefficient result for the intensity used for this study.

[L.125] How did you considered the wind effect during rainfall simulations? Is there any protection?

Reply: We did not put any specific protection against the wind because this system is standing in an open space without a shade but within the closed walls of the department. All the simulations were performed while keeping that in mind that there should minimum influence of wind.

[L.125] Is there any change of the soil surface after each experiment? Is there any change of the roughness, for example?

**Reply:** Thank you for your suggestion. We did not analyze the changes in soil surface after each run for this study, but we will keep that in mind for further studies.**

[L.141] What is the mean intensity of the simulated rainfall? Why are you simulating this intensity?

**Reply: The mean intensity of the simulated rainfall is 36 mm/h.**

[L.147] How did you choose the rainstorm movement velocity?

**Reply: We have followed the studies of De Lima and Singh (2002) and De Lima and Singh (2003) for selecting rainfall movement velocities.**

[L.133] Please change "strom" to storm.

**Reply: Changed.**

[L.288]. check the reference: The style is not correct.

**Reply: Corrected.**

**Reply to Reviewer 2**

This is a review report for the manuscript entitled, "Innovatory rainfall simulator design – A concept of moving storm automation" by Dr. Meena et al. This study proposed a designation of rainfall simulator with a AM+BM controller to simulate moving storm conditions. Later on, they used the rainfall simulator with different moving storms to evaluate the effect of storm movements on time to peak (tp) and peak discharge (Qp) of surface and subsurface flow as well as recession slope, respectively. But, I could not find the result of subsurface flow in either the section of result or discussion. Finally, the authors proposed a multiple regression model to estimate the tp and Qp under different conditions. Three independent variables, saying direction, hillslope gradient, and velocity of storm movement were taken into consideration.

Reply: We are thankful to the reviewer for reviewing our manuscript and providing constructive comments. This paper aims to develop a mobile operated programmable moving-storm rainfall simulator and verify the functionality of developed RS by conducting multiple test runs in different velocities and slope conditions. In the current study, we only used the surface flow component of soil flume. However, the sub-surface component of soil flume can be used in future studies.

In general, the study is a good technical note with a preliminary test rather than an article. The structure is well-organized and the writing is good and clear; however, the findings are expected. The effect of storm movement on hydrograph in terms of tp and Qp, in fact, is associated with relativity (or the tension). To deal with this relativity relies on the competition between runoff velocity (including slope, surface roughness, and slope length) and the velocity of storm movement. Only when the difference of the two velocities is large enough, the hydrograph would be changed. Otherwise, the change in hydrograph could not be detected. Certainly, the high recording frequency can help to describe the hydrograph change.

Reply: We agreed with the reviewer's observation. As mentioned previously, this study developed a moving storm rainfall simulator and verified its functionality by conducting rainfall-runoff experiments. Thus, the obtained results are similar to past studies discussed in the manuscript. Also, the rainfall-runoff experiments were recorded at a temporal frequency of 1m.

Two studies I listed below may help to deal with this issue. What I can suggest for this study is to replace the multiple regression with a conceptual framework to express the relativity issue. Also, two or three additional sets of experiments are encouraged to investigate the effect of storm movement on surface and subsurface flow.

Huang, J.C., Yu, C.K.\*, Lee, J.Y., Cheng, L.W., Lee, T.Y., Kao, S.J. (2012) Linking typhoon tracks and spatial rainfall patterns for improving flood lead time predictions over a mesoscale mountainous watershed, Water Resources Research, 48: W09540, doi:10.1029/2011WR011508.

Huang, J.C.\*, Kao, S.J., Lin, C.Y., Chang, P.L., Lee, T.Y., Li, M.H. (2011) Effect of subsampling tropical cyclone rainfall on flood hydrograph response in a subtropical mountainous catchment, Journal of Hydrology, 409 (1-2): 248-261, doi: 10.1016/j.jhydrol. 2011.08.037.

Citation: https://doi.org/10.5194/hess-2021-502-RC2

Reply: We are thankful to the reviewer for suggesting these papers. The data used in abovementioned studies were collected using radar. Typhoon data is too dynamic and complex to understand.

As per the results of our study, the developed RS can be used to simulate a dynamic storm to study the complex phenomenon under controlled conditions. We used a regression model to explain the obtained results. However, the conceptual framework can be used in future studies to describe the outcomes of rainfall simulator experiments. The scope of this study is limited to the initial test for the moving storm RS capabilities.